# Re-Evaluating the Relevance of the Oxygen–Glucose Deprivation Model in Ischemic Stroke: The Example of Cdk Inhibition

**DOI:** 10.3390/ijms24087009

**Published:** 2023-04-10

**Authors:** Tine D’aes, Quentin Marlier, Sébastien Verteneuil, Pascale Quatresooz, Renaud Vandenbosch, Brigitte Malgrange

**Affiliations:** 1Laboratory of Developmental Neurobiology, GIGA-Stem Cells & GIGA-Neurosciences, University of Liège, 4000 Liège, Belgium; tine.daes@uliege.be (T.D.); q.marlier@dendrogenix.com (Q.M.); sebastien.verteneuil@uliege.be (S.V.); rvandenbosch@uliege.be (R.V.); 2Dendrogenix, Avenue de l’Hôpital, 1—B34 +3, 4000 Liège, Belgium; 3Division of Histology, Department of Biomedical and Preclinical Sciences, University of Liège, 4000 Liège, Belgium; pascale.quatresooz@uliege.be

**Keywords:** neurons, cell culture, OGD, cortex, mouse, cell culture

## Abstract

Previous research has shown that cyclin-dependent kinases (Cdks) that play physiological roles in cell cycle regulation become activated in post-mitotic neurons after ischemic stroke, resulting in apoptotic neuronal death. In this article, we report our results using the widely used oxygen–glucose deprivation (OGD) in vitro model of ischemic stroke on primary mouse cortical neurons to investigate whether Cdk7, as part of the Cdk-activating kinase (CAK) complex that activates cell cycle Cdks, might be a regulator of ischemic neuronal death and may potentially constitute a therapeutic target for neuroprotection. We found no evidence of neuroprotection with either pharmacological or genetic invalidation of Cdk7. Despite the well-established idea that apoptosis contributes to cell death in the ischemic penumbra, we also found no evidence of apoptosis in the OGD model. This could explain the absence of neuroprotection following Cdk7 invalidation in this model. Neurons exposed to OGD seem predisposed to die in an NMDA receptor-dependent manner that could not be prevented further downstream. Given the direct exposure of neurons to anoxia or severe hypoxia, it is questionable how relevant OGD is for modeling the ischemic penumbra. Due to remaining uncertainties about cell death after OGD, caution is warranted when using this in vitro model to identify new stroke therapies.

## 1. Introduction

Stroke is the second leading cause of death worldwide and a primary cause of adult disability, and therefore has enormous societal and personal costs. Most strokes (around 85%) are ischemic and caused by a blockage of a blood vessel to the brain, as opposed to hemorrhagic strokes caused by bleeding. In both cases, disruption of the blood flow to the brain results in a lack of oxygen and glucose, leading to rapid energy failure and death of brain cells.

It is commonly accepted that cells in the ischemic core, where ischemia is most severe, die within minutes due to necrotic death [1]. Since brain cells depend almost exclusively on oxidative phosphorylation as their energy source, a lack of oxygen and glucose due to disruption of the blood flow leads to breakdown in the production of ATP. Consequently, ATP-dependent ion pumps on cell membranes fail, leading to cell swelling and rapid necrosis [2]. Surrounding the ischemic core is a region of stressed but viable cells called the penumbra. The cells in the penumbra region are in a state of metabolic stress and may be functioning at a reduced capacity due to the decreased blood flow and oxygen supply. However, they are not necessarily irreversibly damaged like the cells in the ischemic core. If blood flow is restored quickly enough, the cells in the penumbra region might recover and resume normal function [1]. Despite decades of intensive research, current therapeutic options are still limited to pharmacological or surgical reperfusion therapy to restore blood flow, while no neuroprotective or regenerative drugs for ischemic stroke exist to date. Importantly, reperfusion also causes damage, leading to ischemia-reperfusion injury (IRI).

Since brain damage in the case of ischemic stroke is mainly triggered by a lack of oxygen and glucose, a very straightforward way to model this in vitro is to deprive neural cells or organotypic brain slices of oxygen and glucose by incubating them in a glucose-free medium under hypoxic or anoxic circumstances. After a specified time, cultures can be moved back to normal culture conditions, causing IRI. Oxygen–glucose deprivation (OGD) is the most used in vitro model of ischemic stroke. However, it is not entirely clear how well OGD models real-life stroke, and literature reports on the modes of cell death after OGD are conflicting.

Cell death has classically been categorized as type I (apoptosis), type II (autophagy), or type III (necrosis), among which apoptosis has traditionally been defined as programmed cell death, in sharp contrast with the necrotic explosion in response to overwhelming injury [3,4]. As previously mentioned, it is well-established that cell death in the ischemic core is necrotic, while the ways in which cells die in the penumbra remain unclear and might involve all three major types of cell death. In addition to apoptosis and autophagy, various types of necrosis have been observed after ischemia in vivo, such as necroptosis, ferroptosis, and parthanatos [5]. Interestingly, glutamate excitotoxicity, one of the primary mechanisms in ischemic neuronal death, may lead to all these types of cell death [6]. In ischemic stroke, reduced oxygen delivery leads to a drop in ATP production. This leads to failure of the Na^+^/K^+^ ion pumps that maintain resting potential, and subsequently to an ion imbalance that results in less negative charge inside neurons (i.e., depolarization). Depolarization results in glutamate release, which is exacerbated by impaired reuptake of glutamate by astrocytes. Excessive glutamate activates post-synaptic NMDA receptors that act as Ca^2+^ ion channels. Increased intracellular Ca^2+^ finally activates several programs of neuronal death [7].

A surprising group of proteins implicated in ischemic neuronal death is cyclin-dependent kinases (Cdks), a large family of serine/threonine protein kinases whose activity depends on regulatory subunits called cyclins. Cdks play roles in cell cycle regulation and transcription [8]. Cdks 1, 2, 4, and 6 are mainly expressed in dividing cells to regulate cell cycle progression with their associated cyclins. A unique position is taken by Cdk7, which has roles in transcription as part of the transcription factor II H, as well as in the cell cycle. Cdk7 forms trimeric Cdk-activating kinase (CAK) complexes with the regulatory subunit cyclin H and the assembly factor Mat1. Cdk7 is required for the activating phosphorylation of a threonine residue in the activating segment (T-loop) of cell cycle Cdks 1, 2, 4, and 6 [9,10]. Terminally differentiated neurons withdraw from the cell cycle and cease to divide, which is accompanied by the downregulation of cell cycle Cdks [11]. Interestingly, research has shown that aberrant activation of cell cycle Cdks in post-mitotic neurons leads to cell death rather than division [12]. Previous studies have shown that cell cycle Cdks may become inappropriately expressed and activated in ischemic neurons [13] and are essential mediators of apoptotic neuronal death after ischemia [11]. Preclinical studies have shown reduced neuronal death, smaller infarct size, and improved neurological outcomes using broad-range Cdk inhibitors such as flavopiridol [14], olomoucine [15], and roscovitine [16]. While all these molecules inhibit (at least partially) Cdk7, no studies have investigated the potential involvement of Cdk7 in brain ischemia. Thus, we hypothesized that Cdk7, as part of the CAK complex, might act as an upstream regulator of Cdk activation in neurons following ischemic insult.

In the present work, we investigated the role of Cdk7 in OGD-induced neuronal death on primary cortical mouse neurons. We did not find any evidence of neuroprotection when inhibiting Cdk7 or its targets, the cell cycle Cdks. Despite the well-established idea that apoptosis contributes to cell death in the ischemic penumbra, we also found no evidence of apoptosis in the OGD model. Cortical neurons exposed to OGD appeared predisposed to die in a glutamate-dependent manner that is not easily rescued by inhibiting downstream executioners of programmed cell death. This may match the idea that neurons become “determined to die” upon exposure to excitotoxicity [17]. Given the lack of apoptosis, it also raises questions on how well the OGD models the ischemic penumbra.

## 2. Results

### 2.1. Inhibition or Genetic Deletion of Cdk7 Does Not Improve the Survival of Primary Cortical Neurons Exposed to OGD

Primary cortical neurons were exposed to 4 h of OGD treatment (Figure 1A). To investigate whether Cdk7 plays a role in ischemic neuronal death, cortical neurons were then incubated during OGD either with the Cdk7 specific inhibitor YKL-5-124, the caspase inhibitor Q-VD-OPh (to prevent apoptotic cell death), the NMDA receptor antagonist MK801 (which was used as a positive neuroprotective control), or vehicle (0.1% DMSO). After 24 h of reperfusion, we examined cell viability using a WST-8 assay (CCK8). Cytotoxicity was assessed with a lactate dehydrogenase (LDH) assay, which is a measure of the activity of LDH that is released into the cell culture medium upon cell death, and it is therefore proportionate to the number of dead cells. Upon OGD exposure, cell viability dropped significantly to 38.33% of that of cultures that were kept under atmospheric oxygen levels and in medium with glucose (“normoxia”) for the same duration (Figure 1B). Neither the Cdk7 inhibitor YKL-5-124 nor the caspase inhibitor Q-VD-OPh counterbalanced this deleterious effect on neuronal viability. Only the positive control MK801 significantly improved neuronal viability after 4 h of OGD to 76.03% of normoxia levels (*p* = 0.0002) and significantly decreased LDH activity (*p* = 0.0113) (Figure 1B). In addition to these quantitative measures, neuronal morphology was visualized using immunostainings for β-tubulin III (Tuj1). This revealed that neurites are largely destroyed after 4 h of OGD. Incubating neurons with MK801 allowed us to partially preserve the neurite structure (Figure 1C). Such protection was not observed with Cdk inhibitors.

To confirm the results obtained with the Cdk7 inhibitor YKL-5-124, we genetically invalidated Cdk7 in post-mitotic neurons using the Cre–lox system. We isolated primary cortical neurons from embryos harboring a floxed Cdk7 allele (Cdk7^lox^) and expressing the Cre recombinase in all post-mitotic pyramidal neurons (Nex^Cre^). Cdk7^lox/lox^ Nex^+/+^ (control) and Cdk7^lox/lox^ Nex^Cre/+^ (Cdk7 KO) neurons were cultured and exposed to 4 h of OGD. Successful deletion of Cdk7 was confirmed using immunofluorescence staining and Western blot. In cortical cultures obtained from Cdk7-KO mice, Cdk7 levels were significantly decreased to 28.81% of control cultures (*p* = 0.0078; Figure 2A). The remaining Cdk7 expression in Nex-Cdk7-KO cultures (Figure 2B) was likely attributable to either non-Nex expressing neurons (GABAergic neurons) or contaminating glial cells in the culture. Genetic deletion of Cdk7 did not improve the survival of primary cortical neurons exposed to OGD (Figure 2C), validating the results obtained with YKL-5-124.

### 2.2. Pharmacological Inhibition of Cdk7 Targets Does Not Improve the Survival of Primary Neurons Exposed to OGD

Cdk7 is responsible for the phosphorylation and activation of Cdk1, Cdk2, and Cdk4/6. To investigate if inhibition of these cell cycle Cdks might be neuroprotective in the OGD model, we tested different inhibitors, including RO-3306, a specific Cdk1/Cdk2 inhibitor, palbociclib, a specific Cdk4/Cdk6 inhibitor, and roscovitine, which is known to target Cdk1, 2, 5, 7, and 9. None of these inhibitors provided neuroprotective effects following 4 h of OGD (Figure 3A).

### 2.3. Apoptosis Does Not Play a Significant Role in Neuronal Death in the OGD Model

Since caspase activation requires ATP, apoptosis may not occur in the case of severe ATP depletion. Therefore, we reduced OGD duration to 2 h to investigate whether this would lead to more apoptosis and, thus, neuroprotection with anti-apoptotic drugs. Upon shorter OGD, MK801 was still the only significantly neuroprotective drug (Figure 3B).

Considering that Cdk-related neuronal death is supposed to be apoptotic, we next sought to investigate whether any apoptosis occurred in our OGD model. Therefore, we performed immunofluorescence stainings for cleaved caspase-3 (CC3) at different time points after OGD (Figure 4A). The percentage of CC3^+^ cells among the total population at each time point after OGD was not statistically different from cultures that were kept under normoxia conditions for the same duration (Figure 4B). Surprisingly, we observed an important expression of CC3 in normoxic conditions. This might be explained by apoptosis in the cell culture or by physiological functions of CC3 as recently described [18,19].

To ensure that the CC3 staining was associated with apoptosis and to test whether Q-VD-OPh was used at an efficient concentration, we used staurosporine (100 nM), a robust inducer of apoptosis. As expected, staurosporine significantly increased the number of CC3^+^ cells (Figure 4C). When staurosporine was combined with Q-VD-OPh, the percentage of CC3^+^ cells was significantly decreased compared to staurosporine alone (Figure 4D). These results showed that apoptosis could be induced in our cell culture model and delayed using a pan-caspase inhibitor. In addition, it implies that cortical neurons die in a caspase-independent manner in the OGD model.

### 2.4. Excitotoxicity Plays a Crucial Role in the OGD Model, but Inhibiting Downstream Cell Death Pathways Does Not Rescue Neuronal Death

To further explore the mechanisms involved in neuronal cell death after OGD, we evaluated the effect of specific inhibitors on the main classes of cell death types. We tested whether 3-MA, an autophagy inhibitor, and necrostatin-1, an inhibitor of necroptosis, might be protecting neurons after OGD (Figure 5B). Neither drug showed significant neuroprotection (Figure 5A). In addition, a combination of 3-MA, Necrostatin-1, and Q-VD-OPh did not result in significant neuroprotection either (average viability 41.53% of normoxia, compared to 38.33% for the vehicle). This indicated that inhibiting executioners of programmed cell death downstream from NMDA receptor-mediated excitotoxicity did not decrease the number of neurons dying in response to OGD.

## 3. Discussion

Despite the well-established idea that apoptosis contributes to cell death after ischemic stroke, we found no evidence of apoptosis in the OGD model we used. There was no increase in the number of cells that expressed CC3 in primary murine neurons exposed to OGD. In addition, we could not prevent neuronal death with pan-caspase inhibition. This is likely due to various factors, including the specific cell type being studied, the duration and severity of the OGD exposure, and the outcome measures to assess cell survival and cell function. 

Our study used mouse cortical neurons at E15.5 cultured for 5DIV before OGD exposure. This relatively short culture period avoids giving astrocytes enough time to grow and proliferate. In cortical cultures, neurons and astrocytes are first seen after 1 DIV and 5–6 DIV, respectively, and the number of astrocytes in the culture then increases between 5 DIV and 9 DIV [20]. Conditioned medium from astrocytes exposed to OGD augments apoptosis in cultured cortical neurons [21], suggesting that the low number of astrocytes in 5 DIV cultures may prevent neuronal apoptosis. In accordance with this, Jones et al. detected no apoptosis in cortical cultures treated beforehand with cytosine arabinoside (AraC) to eliminate astrocytes [22]. Therefore, an interesting future experiment could consist of adding a conditioned medium from astrocytes to the neuronal culture to see whether this affects our results. Some studies have suggested that certain types of cells may be more resistant to OGD-induced apoptosis due to their ability to upregulate protective genes and produce antioxidants that can help mitigate cellular damage. Neural precursor cells may still be present in our culture conditions, as previously shown [20]. These cells are intrinsically more resistant than neurons to OGD [23] and could contribute to the lack of apoptosis we observe in our model.

Researchers often use flow cytometry for Annexin V (combined or not with 7AAD), TUNEL staining, CC3 Western blots, and CC3 activity assays to assess apoptosis. Our study used CC3 immunostainings that may not be sensitive enough. However, as previously described, we detected increased apoptosis in the presence of staurosporine [24]. In addition, the pan-caspase inhibitor Q-VD-OPh did not improve cell survival 24 h post-OGD, although it was able to reduce staurosporine-induced apoptosis. Altogether, these data indicate that the contribution of apoptosis is low, if not absent, in our culture conditions. Finally, variability in apoptotic results could be due to a difference in the duration of OGD exposure. This can lead to variations in the degree of cellular stress and the protective mechanisms activated. Even though many articles report apoptotic death of primary neurons in vitro after OGD, these claims are sometimes merely based on the TUNEL assay, a measure of DNA fragmentation, which is not specific for apoptosis [25,26]. In addition, most papers use (cortical) neurons from rats rather than mice, suggesting that there might be species-related differences. Others, like us, observed no evidence of apoptosis after OGD and have found that neuronal death in the OGD model is mostly necrotic [27]. The lack of consensus on the neuronal cell death type after OGD is food for thought. Since the OGD model is used as a screening tool to identify new stroke treatments, it should reflect cell death mechanisms that actually take place in in vivo stroke.

To further clarify the nature of neuronal death in the OGD model we used, we also tested neuronal survival after OGD in the presence of inhibitors of two other major types of cell death: autophagy and necroptosis. Although apoptosis as programmed cell death has traditionally been defined in sharp contrast with the passive necrotic explosion in response to overwhelming injury, programmed types of necrosis such as necroptosis share several key mechanisms with apoptosis and can be triggered by the same death signals [4]. In fact, necroptosis was first described as caspase-independent cell death triggered by exposure to TNF, that exclusively occurs in the presence of a pan-caspase inhibitor such as Q-VD-OPh [28,29]. Treatment of our neuronal cultures with necrostatin-1, a specific necroptosis inhibitor [30], did not affect cell death following OGD. This is in contrast with in vivo findings showing a neuroprotective effect of necrostatin-1 in the middle cerebral artery occlusion (MCAO) model [31]. Since necroptosis is mostly triggered by proinflammatory molecules such as TNF secreted by glial cells, not much necroptosis might be expected in isolated neuronal cultures. Regarding autophagy, studies have shown that the effects of 3-MA on OGD-induced cell death can vary depending on the experimental conditions. Some studies have reported that inhibition of autophagy with 3-MA leads to increased cell death during OGD, while others have reported no effect or even protective effects [4]. Our study did not find any protective effect of 3-MA against OGD-induced neuronal death. These discrepancies may be due to differences in the duration and severity of OGD, the cell type used, and the concentration and duration of 3-MA treatment. We did not investigate whether neurons expressed markers of autophagy and necroptosis, nor did we confirm the efficacy of the autophagy and necroptosis inhibitors as we did for apoptosis, which are limitations of our study.

Whereas we did not demonstrate a better survival of neurons using pharmacological inhibitors of apoptosis, autophagy and necroptosis, the NMDA receptor antagonist MK801 was reproducibly protective in our model. We can therefore conclude that embryonic cortical neuronal cultures under OGD conditions die in a manner that relies heavily on the NMDA glutamate receptor, mimicking the “excitotoxic programmed necrosis” that rapidly occurs after cerebral ischemia and other types of acute brain injuries, such as prolonged epileptic seizures or brain trauma. This “excitotoxic programmed necrosis” is caspase-independent and requires activation of Ca^2+^-dependent enzymes such as calpain I and neuronal nitric oxide synthase (nNOS) (reviewed in [32]). Future studies should confirm the necrotic nature of this OGD model by measuring the levels of necrotic cell death markers such as the miR-122 [33].

Pharmacological inhibitors of Cdks have been investigated for their potential neuroprotective properties. Cdks are critical in regulating the cell cycle and are involved in other cellular processes, including neuronal differentiation, synaptic plasticity, and neuronal survival [12]. Studies have suggested that Cdk inhibitors may be able to protect neurons from various forms of damage and degeneration. While they were originally designed to target specific Cdks, such as Cdk1/2 or Cdk4/6, they possess off-target activities on other Cdks and other families of kinases. To specifically address the role Cdk7 in our OGD model, we took advantage of YKL-5-124, a recently developed Cdk7 inhibitor, that forms a covalent bond with a conserved cysteine residue (Cys312) located outside the kinase domain [34]. This irreversible inhibition is thought to contribute to the potency and selectivity of this molecule. In contrast to neurons treated with MK801, we could not demonstrate any protection with YKL-5-124, suggesting that Cdk7 is not involved in OGD-induced neuronal death. The lack of neuroprotective effects observed with YKL-5-14 could be due to several factors. One possible explanation is that Cdk7 plays a complex and multifaceted role in the nervous system. Depending on the context and cell type, its inhibition may have beneficial and detrimental effects. While Cdk7 inhibition has been shown to have anti-cancer effects by blocking cell cycle progression and inducing apoptosis [35,36], it may also negatively affect normal cells and tissues. In particular, Cdk7 plays a role in transcriptional regulation and gene expression, and its inhibition may disrupt normal gene expression patterns and interfere with critical neuronal or glial processes (even if inhibition of Cdk7 alone may not affect general transcription) [33,37]. It is also worth mentioning that while Cdk7 functions as a CAK in dividing cells, this is not necessarily the case in post-mitotic neurons, which may explain the absence of neuroprotection observed with YKL-5-124. For instance, JNKs have been identified as CAK for Cdk4 in cancer cells [38], but such regulation has not yet been studied in neurons. Further research is needed to understand how Cdks are regulated in neurons and determine their downstream targets.

To conclude, our results observed with the OGD model contribute to the incongruent literature on the mode of cell death that is responsible for OGD-induced neuronal death. They suggest that this model should be adapted to be more reproducible and to better model delayed neuronal death as it occurs in the ischemic penumbra, whether this is truly through apoptotic pathways or not. Not only will it contribute to a better understanding of the pathophysiological processes in the penumbra, but it will also help the scientific community compare results, especially when studying the effects of drugs that target the same molecular cascades.

## 4. Materials and Methods

### 4.1. Mice

Wild-type pregnant CD1 mice were obtained from the Animal Facility of the University of Liège. To genetically invalidate Cdk7 in glutamatergic neurons, primary cortical neurons were cultured from Cdk7^lox/lox^ mice [10] that were crossed with Nex^Cre/+^ mice [39] to obtain Cdk7^lox/lox^ Nex^Cre/+^ animals (Nex-Cdk7-KO) in a mixed C57BL/6J and 129/Sv background. Primary cortical neurons from Cdk7^lox/lox^ Nex^+/+^ littermates were used as controls. Mice were maintained in facilities where the temperature was kept constant at 22 °C. Animals were subjected to a 12 h/12 h day/night cycle and had unrestricted access to food and water. The ethics committee of the University of Liège approved all experimental procedures for animal experiments (ethical dossier n°19-2129).

### 4.2. Primary Cortical Neuronal Cultures

Cortices from E15.5 pregnant female mice were microdissected in PBS-glucose on ice, incubated for 20 min at 37 °C in a solution of 0.25% trypsin and 0.01% DNase (0.01%) in EBSS, and incubated in FBS (Sigma, F7524, St. Louis, MO, USA) for 10 min at room temperature. Cortices were then rinsed with neurobasal medium (Gibco, Billings, MO, USA, 21103049) and suspended in neurobasal medium supplemented with 2% B27 (Gibco, 17504044), 2 mM L-glutamine (Lonza, BE17-605E, Rockville, MD, USA), and 1% penicillin-streptomycin mixture (100 U/mL–100 µg/mL; Lonza, DE17-602E, Rockville, MD, USA) (hereafter called supplemented neurobasal medium). After dissociating the tissue with a glass pipette, the cell suspension was filtered through a 70 µm filter. Cell concentration was determined, and cells were plated at a density of 300,000 cells/mL in multi-well plates coated with 40 µg/mL poly-D-lysine (Sigma, A-003-E) and 6 µg/mL laminin (Sigma, L2020). Three hours after plating, half of the culture medium was replaced with fresh supplemented neurobasal medium. Primary neurons were maintained in culture for five days in an incubator with 5% CO_2_ and 95% air at 37 °C before further experiments.

### 4.3. OGD

For OGD exposure, supplemented neurobasal medium was replaced with DMEM without glucose (Gibco, 11966025), and cortical neurons were transferred to an incubator with 1% O_2_, 5% CO_2_, and 94% N_2_. Control neurons were kept in supplemented neurobasal medium in an incubator with 5% CO_2_ and 95% air for the same period. After four (or two) hours, primary neurons were placed back in complete neurobasal medium and maintained in a standard CO_2_ incubator with 5% CO_2_ and 95% air for 24 h (Figure 1A). Control and OGD-exposed cells were incubated either with an inhibitor or 0.1% DMSO. Inhibitors were MK801 (Tocris, 0924, Ellisville, MO, USA), Q-VD-OPh (Sigma, SML0063), YKL-5-124 (MedChemExpress, HY-101357B, Princeton, NJ, USA), roscovitine (Selleckchem, S1153, Houston, TX, USA), palbociclib (Selleckchem, S1116), RO-3306 (AdipoGen, AG-CR1-3515-M005, San Diego, CA, USA), 3-methyladenine (3-MA; Selleckchem, S2767), or necrostatin-1 (AdipoGen, AG-CR1-2900-M005). After 2 or 4 h, cells were placed back in the complete neurobasal medium and maintained in a standard CO2 incubator with 5% CO2 and 95% air for up to 24 h. Each experiment on wild-type neurons was replicated at least four times, meaning that at least four independently obtained batches of primary cortical neurons were used, each obtained from the embryos of a different pregnant mouse. Every biological replicate (corresponding to one dot on the graph) in turn consisted of four technical replicates (four separate wells with cells from the same batch).

### 4.4. Viability and Cytotoxicity Assays

Cell viability was assessed using a cell counting kit-8 (CCK8) assay (MedChemExpress, HY-K0301). For this, 10 µL of CCK8 solution was added per well of a 96-well plate 24 h after the end of OGD, and cells were incubated at 37 °C for 3 h. Absorbance was measured at 450 nm using a Thermo Labsystems Multiskan Ascent microplate reader.

Cytotoxicity was determined using a lactate dehydrogenase (LDH) assay from Promega (G1780) according to the manufacturer’s instructions. Briefly, 24 h after the end of OGD or immediately after 24 h of incubation with pharmacological inhibitors, 50 µL of cell culture supernatant was collected from each well, and an equal volume of CytoTox 96 reagent was added. After 30 min, 50 µL of stop solution was added, and absorbance was measured using a Thermo Labsystems Multiskan Ascent microplate reader.

### 4.5. Staurosporine Treatment

Primary neurons were incubated with 100 nM of staurosporine (AG-CN2-0022 AdipoGen Life Sciences) for 24 h to induce apoptosis. Immediately after the 24 h of incubation, cells were fixed for immunostaining.

### 4.6. Immunofluorescence Stainings

For all IF stainings, primary neurons on coverslips were fixed for 15 min in 4% paraformaldehyde (PFA) at room temperature. Cells were permeabilised and blocked in PBS containing 10% donkey serum (SouthernBiotech, 0030-01) and 0,1% Triton X-100 for (108603 Merck Millipore) for 1 h at room temperature during gentle shaking. Incubation with primary antibodies in PBS-Triton 0.1% with 5% donkey serum was performed overnight at 4 °C in the following dilutions: mouse monoclonal anti-β-III tubulin (Tuj1) (1:1000; BioLegend # 801202), rabbit anti-cleaved caspase-3 (1:1000; Cell Signaling Technology # 9661), and mouse anti-Cdk7 (1:100, Santa Cruz # sc-7344). After three washing steps with PBS, secondary antibodies (Alexa Fluor^TM^, Invitrogen) were used 1:1000 in PBS with 5% donkey serum and 0.1% Triton X-100 for 1 h at room temperature. Cells were washed two more times with PBS, counterstained for 10 min with DAPI 1:5000 (D9542 Sigma), and mounted with fluorescence mounting medium (Dako, S3023) before confocal microscopy (Olympus FV1000).

### 4.7. Western Blots

Pellets of primary neurons from either Cdk7^L/L^ Nex^Cre/+^ or Cdk7^L/L^ Nex^+/+^ mice were incubated in extraction buffer, a mixture (pH 8) of 150 mM NaCl (Roth, 3957.1), 50 mM tris-HCl (Roth, 9090.3), 60 mM β-glycerophosphate (Sigma, G9422), 10 mM sodium phenyl phosphate dibasic dihydrate (Sigma, P7751), 500 mM NaF (Sigma, S-7920), 0.2 mM sodium orthovanadate (Sigma, S-6588), and 1% NP40 (I8896 Sigma) with protease inhibitors Complete Mini EDTA-Free (Roche, 11836170001). Homogenized cells in extraction buffer were left on ice for 20 min and centrifuged for 15 min (15,000× *g*) at 4 °C to obtain whole cell lysates. The supernatant was collected, and protein concentration was determined using a Pierce^TM^ bicinchoninic acid (BCA) protein assay kit (Thermo Scientific, 23225) according to the manufacturer’s instructions. Before migration, samples were denatured for 2 min at 95 °C. Next, a protein ladder (Thermo Scientific, 26616) and 15–30 µg of protein in Laemmli buffer were loaded onto the gel. Proteins were allowed to migrate on 4–12% Bis-Tris gels (Invitrogen, NW04120BOX) immersed in MES migration buffer (Novex, B0002) for around 1.5 h at 165 V. Proteins were then transferred to a polyvinylidene fluoride (PVDF) membrane (Millipore, IPVH00010) at 300 mA in transfer buffer (25 mM Tris with 192 mM glycine (Roth, 3908.2) in 20% methanol (ChemSolute, 1437.2511) and 80% distilled water). Membranes were blocked for 1 h at RT in tris-buffered saline with Tween 20 (TBST; VWR, M147) and 5% milk (Roth, T145.3). Next, the membrane was incubated in the same solution (TBST with 5% milk) with the primary antibody (mouse anti-Cdk7, 1:500, Santa Cruz # sc-7344) overnight at 4 °C. The following day, after washing three times with TBST, the membrane was incubated for 1 h at RT with the secondary antibody conjugated to HRP (anti-mouse (Invitrogen, G21040; 1:10000) or anti-rabbit (Invitrogen, G21234; 1:5000)). After three final washing steps with TBST, the proteins were revealed through chemiluminescence using an ECL detection kit (Thermo Scientific, Pierce ECL Western Blotting Substrate, 32106) on an Amersham ImageQuant 800 (Cytiva) imaging system. The density of the bands was measured using FIJI/ImageJ software, and relative protein levels were normalized to β-actin levels (actin-HRP 1:25000, Sigma, A3854).

### 4.8. Statistical Analysis

Statistical analysis was performed using GraphPad Prism 8.4.3 statistical software. Since sample sizes were too small for equal variances to be assumed, the conditions for parametric testing were not met. Consequently, differences in data were assessed using the non-parametric Mann–Whitney U test (for two independent groups) or the Kruskal–Wallis test (for more than two groups) with Dunn’s multiple comparisons tests. Results were visualized using column scatter plots with bars corresponding to the median and error bars representing the interquartile range. Differences were considered statistically significant if the *p*-value was less than 0.05.

## Figures and Tables

**Figure 1 ijms-24-07009-f001:**
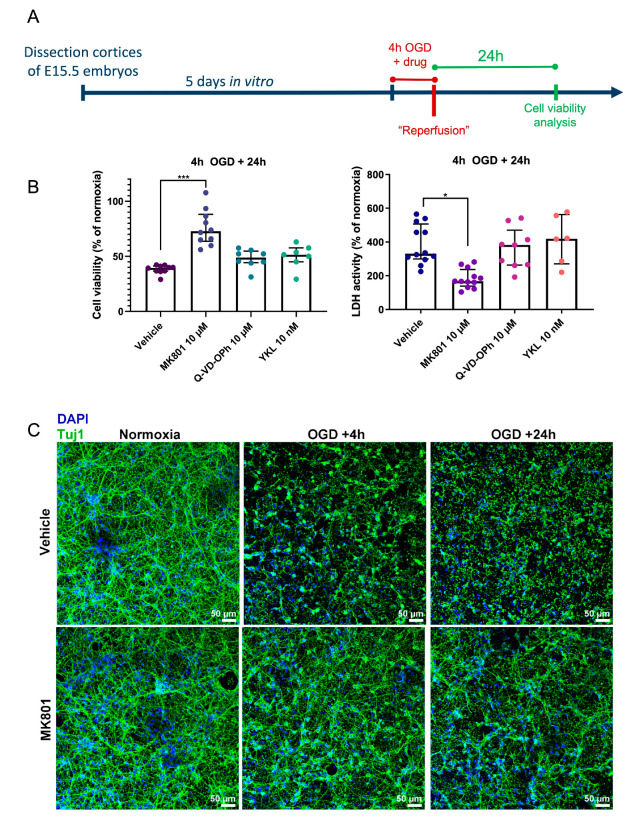
Inhibition of the NMDA receptor protects against ischemic neuronal death, but not inhibition of Cdk7 or caspases. (**A**) Experimental timeline. Primary cortical neurons were obtained from E15.5 mouse embryos. At DIV5, neurons were exposed to 4 h of OGD, while incubated with vehicle (0.1% DMSO) or a specific inhibitor of either Cdk7, the NMDA glutamate receptor, or caspases. Viability and cytotoxicity were assessed 24 h after the end of OGD. (**B**) Viability of primary cortical neurons exposed to 4 h of OGD in the presence of either vehicle only (0.1% DMSO), the NMDA receptor antagonist MK801, the pan-caspase inhibitor Q-VD-OPh, or the Cdk7 inhibitor YKL-5-124 (“YKL”). Neurons were incubated with 0.1% DMSO in all conditions. Viability was measured with the CCK8 assay (left), while cytotoxicity was quantified using the LDH assay (right). Values are expressed as a percentage of normoxia conditions, measured 24 h after the end of OGD. Each dot represents one biological replicate corresponding to a different pregnant mouse, showing the mean value of 4 replicates (4 separate wells) from one batch of primary cortical neurons. Groups were compared using the Kruskal–Wallis test with Dunn’s multiple comparisons test. Error bars represent the interquartile range. * *p* ≤ 0.05; *** *p* ≤ 0.001. (**C**) Representative immunofluorescence images of primary cortical neurons stained for Tuj1 (green) and DAPI (blue) to show the integrity of neurites after OGD in the presence or absence of the NMDA receptor antagonist MK801. Primary cortical neurons were exposed to normoxia or 4 h of OGD in the presence of either 10 µM of MK801 or vehicle (0.1% DMSO). Cells were fixed at either 4 or 24 h after the end of OGD to investigate neurite integrity.

**Figure 2 ijms-24-07009-f002:**
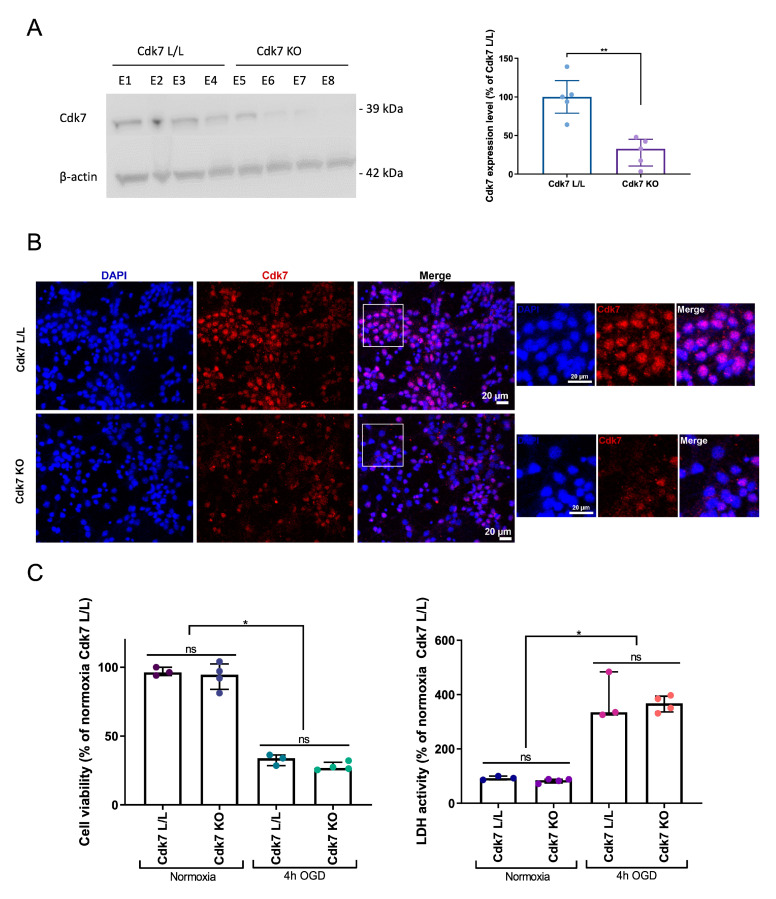
Genetic deletion of Cdk7 does not improve the survival of primary cortical neurons exposed to OGD. (**A**) Western blot showing Cdk7 levels in primary cortical cultures obtained from either Cdk7^lox/lox^ Nex^+/+^ (“Cdk7 L/L”; embryo (E) 1-4) or Cdk7^lox/lox^ Nex^Cre/+^ (“Cdk7 KO”; E5-8) mice. Western blots were quantified using FIJI/ImageJ and normalized to levels of the housekeeping gene β-actin. Protein levels are expressed as % of the mean of the “Cdk7 L/L” condition for that gel. Each dot on the histogram represents a protein band, corresponding to one batch of cortical cells obtained from one embryo. Statistical significance was investigated using the Mann–Whitney test. (**B**) Representative immunofluorescence staining for Cdk7 (red) and counterstaining with DAPI (blue) in primary cortical neurons obtained from either Cdk7^lox/lox^ Nex^+/+^ (“Cdk7 L/L”) or Cdk7^lox/lox^ Nex^Cre/+^ (“Cdk7 KO”) mice. Cdk7 is present in virtually all cells obtained from Cdk7^lox/lox^ mice, while its expression is markedly decreased in the KO condition. (**C**) Viability (CCK8 assay) and cytotoxicity (LDH assay) of primary cortical neurons obtained from either Cdk7^lox/lox^ Nex^+/+^ (“Cdk7 L/L”) or Cdk7^lox/lox^ Nex^Cre/+^ (“Cdk7 KO”) mice were measured 24 h after OGD. Each dot represents the mean value of 4 replicates for one embryo. Groups were compared pairwise using a Kruskal–Wallis test with Dunn’s multiple comparisons test. Error bars: interquartile range. * *p* ≤ 0.05; ** *p* ≤ 0.01; ns = not statistically significant.

**Figure 3 ijms-24-07009-f003:**
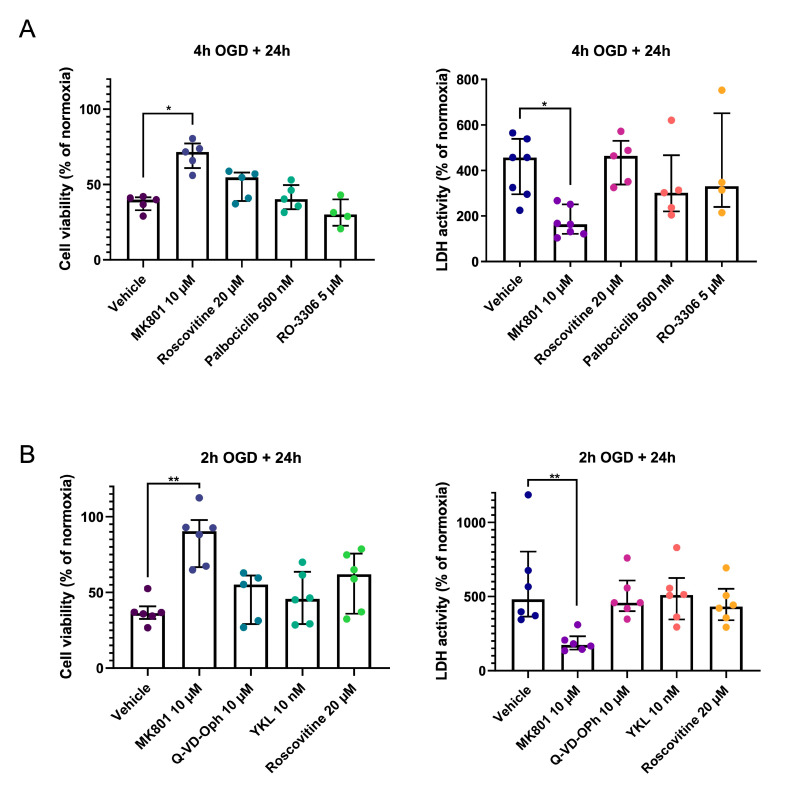
Pharmacological inhibition of Cdk7 targets does not improve neuronal survival after IRI. Results are not changed by shortening the duration of OGD. (**A**) Viability (CCK8 assay) and cytotoxicity (LDH assay) of primary cortical neurons exposed to 4 h of OGD in the presence of either vehicle only (0.1% DMSO), positive control MK801, pan-Cdk inhibitor roscovitine, Cdk4/6 inhibitor Palbociclib, or Cdk1/Cdk2 inhibitor RO-3306. Each dot represents the mean of 4 replicates for one batch of primary cortical neurons (each batch is obtained from a separate pregnant mouse). (**B**) Viability (CCK8 assay) and cytotoxicity (LDH assay) of primary cortical neurons exposed to 2 h of OGD in the presence of either vehicle only (0.1% DMSO), positive control MK801, pan-caspase inhibitor Q-VD-OPh, Cdk7 inhibitor YKL-5-124 (“YKL”), or the broad Cdk inhibitor roscovitine. Each dot represents the mean of 4 replicates for one batch of primary cortical neurons (from a separate pregnant mouse). For all experiments, groups were compared using the Kruskal–Wallis test with Dunn’s multiple comparisons test. Error bars: interquartile range. * *p* ≤ 0.05; ** *p* ≤ 0.01.

**Figure 4 ijms-24-07009-f004:**
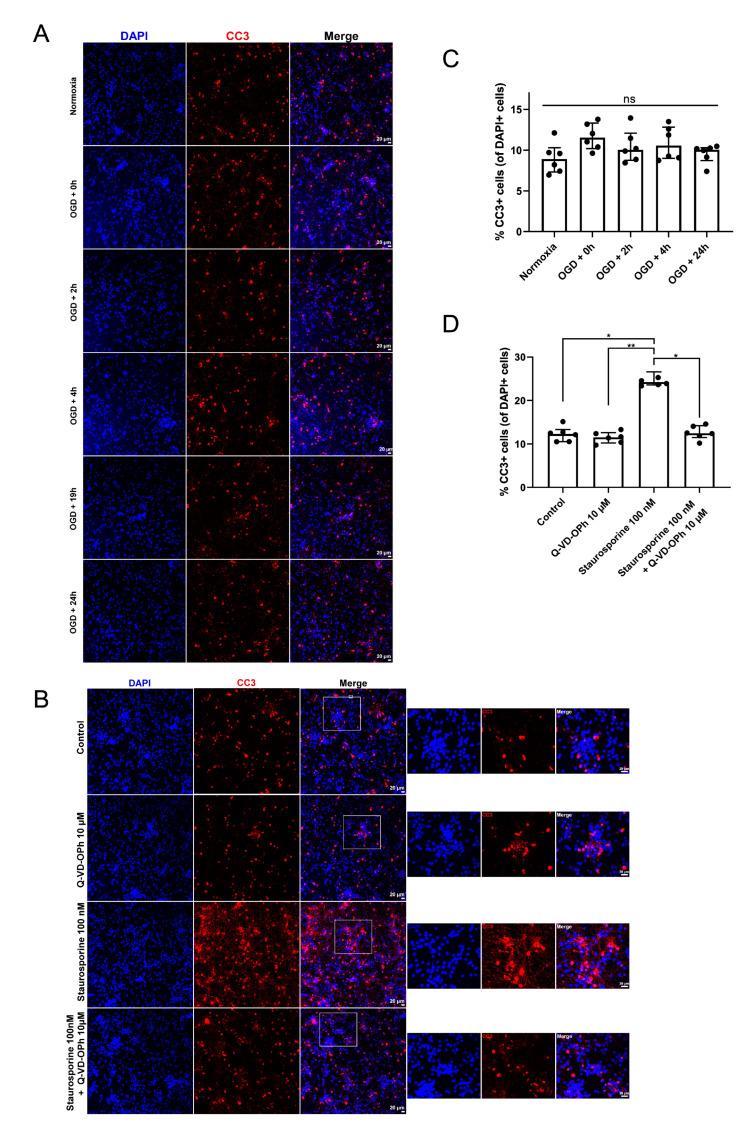
OGD does not result in marked apoptosis of primary cortical neurons. (**A**) Representative immunofluorescence images for cleaved caspase-3 (CC3; red) expression as a readout of apoptosis in primary cortical neurons at several time points after OGD (0–24 h). (**B**) Representative immunofluorescence staining for CC3 expression (red) in primary cortical neurons incubated either with vehicle (0.1% DMSO), 10 µM of the caspase inhibitor Q-VD-OPh, 100 nM of staurosporine, or with both staurosporine and Q-VD-OPh. (**C**,**D**) The number of CC3^+^ cells was counted using FIJI/ImageJ and expressed as a percentage of the total number of cells, determined using DAPI counterstaining (blue). Each dot on the histogram represents the average from 4 different fields of view for one coverslip. Statistical analyses were performed using the Kruskal–Wallis test with Dunn’s multiple comparisons test. Error bars: interquartile range. * *p* ≤ 0.05; ** *p* ≤ 0.01; ns = not statistically significant.

**Figure 5 ijms-24-07009-f005:**
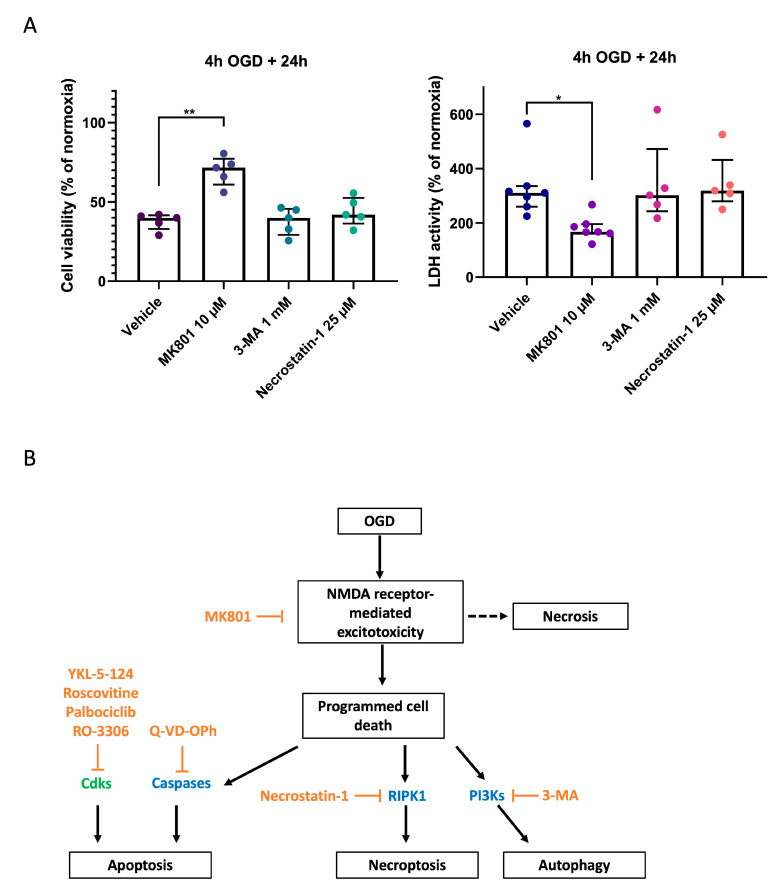
Pharmacological inhibition of autophagy and necroptosis does not prevent neurons from dying in response to OGD. (**A**) Viability of primary cortical neurons exposed to 4 h of OGD in the presence of either vehicle only (0.1% DMSO), MK801 (an NMDA receptor antagonist), 3-MA (an autophagy inhibitor), or necrostatin-1 (an inhibitor of necroptosis). Viability was measured with the CCK8 assay (left), while cytotoxicity was quantified using the LDH assay (right). Values are expressed as a percentage of the normoxia condition, measured 24 h after the end of OGD. Each dot represents the mean value of 4 replicates for a unique batch of primary cortical neurons (each batch is obtained from a separate pregnant mouse). Groups were compared using Kruskal–Wallis test with Dunn’s multiple comparisons test. Error bars: interquartile range. * *p* ≤ 0.05; ** *p* ≤ 0.01. (**B**) Scheme of the hypothetic pathways involved in neuronal cell death following OGD.

## Data Availability

Data are available upon request to the authors.

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
