# Peer review of "Re-Evaluating the Relevance of the Oxygen–Glucose Deprivation Model in Ischemic Stroke: The Example of Cdk Inhibition"

_ijms, 2023, doi:10.3390/ijms24087009_

Round 1
Reviewer 1 Report
D’aes et al. provided data that CDK7 inhibition is not neuroprotective in an OGD model of ischemic stroke. The data is interesting and the manuscript is generally well written. However, several important positive controls are missing from the manuscript and should be addressed before publication in IJMS.
1. The authors inhibited CDK7 using the inhibitor YKL-5-124 at a single concentration (10nM) and did not observe neuroprotective effects after OGD. As a positive control, the authors should show that the inhibitor was effective in inhibiting CDK7, e.g. blotting for CDK7 targets CDK1/2/4/6 T loop phosphorylation or RNA PolII CTD Ser5 phosphorylation. Similar positive controls should be performed for the CDK7 KO experiment and the roscovitine/palbociclib experiments. Dose titration should preferably be done with inhibitor experiments as well.
2. The authors did not observe increased levels of cleaved caspase 3 following OGD. Is CDK activity/cell-cycle re-entry observed following OGD?
3. The authors tested whether 3-MA and necrostatin-1 are neuroprotective following OGD. Again, positive control experiments similar to those in Figure 4D (which is a great experiment) would be helpful in interpreting these results.
4. For replicates, the authors mostly used n = 4 technical replicates from one batch of primary neurons obtained from embryos of one pregnant mouse. The authors should repeat key experiments using true biological replicates, e.g. different batches of neurons from multiple mice.
5. The authors discussed the potential importance of astrocytes in mediating apoptosis after OGD. Have the authors tried CDK inhibition experiments using more mature cortical cultures or applying conditioned medium from astrocytes?
6. Since CDK5 does not require T loop activation by CDK7, could this be an explanation of why flavopiridol, which inhibits CDK5 as well as other CDKs, is neuroprotective but CDK7 inhibition is not?
Author Response
reviewer 1
D’aes et al. provided data that CDK7 inhibition is not neuroprotective in an OGD model of ischemic stroke. The data is interesting and the manuscript is generally well written. However, several important positive controls are missing from the manuscript and should be addressed before publication in IJMS.
- The authors inhibited CDK7 using the inhibitor YKL-5-124 at a single concentration (10nM) and did not observe neuroprotective effects after OGD. As a positive control, the authors should show that the inhibitor was effective in inhibiting CDK7, e.g. blotting for CDK7 targets CDK1/2/4/6 T loop phosphorylation or RNA PolII CTD Ser5 phosphorylation. Similar positive controls should be performed for the CDK7 KO experiment and the roscovitine/palbociclib experiments. Dose titration should preferably be done with inhibitor experiments as well.
We thank the reviewer for this valuable comment. Indeed, we have tried to verify the effective inhibition of Cdk7 with YKL-5-124 through western blots for p-Thr161-Cdk1 but found no effect of Cdk7 inhibition on the phosphorylation of Cdk1. While these results were surprising, we similarly found no decrease in phosphorylated Cdk1 in Cdk7-KO neurons. This has led us to hypothesize that Cdk7 may perhaps not act as CAK in non-dividing neuronal cells, as mentioned in the discussion.
While we also tested p-Thr160-Cdk2 antibodies, these, unfortunately, did not provide reliable results.
In addition, several studies have shown that Cdk7 is not essential for the phosphorylation of RNA pol II, but that Cdk12/13 may compensate for the loss of Cdk7 activity (Olson et al., 2019).
In short, finding an easy readout of Cdk7 inhibition is rather challenging. However, we found that YKL-5-124 clearly inhibited the proliferation of cell lines without producing toxicity, suggesting successful inhibition of Cdk7.
Moreover, we believe that the experiments with the genetic invalidation of Cdk7 are a very strong confirmation of the negative results obtained with YKL-5-124.
Regarding the used concentrations, for YKL-5-124, a concentration was chosen at around the reported IC50 for the CAK complex (9.7 nM; (Olson et al., 2019). We remark that concentrations above 20 nM were toxic for primary neurons and therefore we did not use higher concentrations that 10nM. 3-MA and necrostatin-1 were used at a concentration for which efficacy in preventing autophagic (Li et al., 2014) or necroptotic (Wu et al., 2015) cell death, respectively, has previously been shown in neuronal cells. Two concentrations were tested for each (1 and 2.5 mM for 3-MA and 25 and 50 μM for necrostatin-1), of which only the lower one is shown in Figure 5A since the higher concentration produced similar (negative) results.
- The authors did not observe increased levels of cleaved caspase 3 following OGD. Is CDK activity/cell-cycle re-entry observed following OGD?
In our own experiments, we indeed observed a significant upregulation of Cdk1 following OGD, which is in accordance with results previously published by (Marlier et al., 2018). While Cdk1 upregulation does not necessarily mean cell-cycle re-entry, another study demonstrated that OGD induces S-phase entry (i.e. BrdU incorporation) and hyper-phosphorylation of the retinoblastoma protein(Yu et al., 2012).
- The authors tested whether 3-MA and necrostatin-1 are neuroprotective following OGD. Again, positive control experiments similar to those in Figure 4D (which is a great experiment) would be helpful in interpreting these results.
We agree that experiments inducing autophagy and necroptosis would be great ways to confirm the efficacy of the used drugs. However, we would like to argue that the focus of our experiments was on apoptosis, given the alleged apoptotic nature of Cdk-related neuronal death. We did acknowledge this limitation (hereby also addressing concerns raised by reviewer 3) by adding the following sentence in the discussion:
“We did not investigate whether neurons expressed markers of autophagy and necroptosis, nor did we confirm efficacy of the autophagy and necroptosis inhibitors like we did for apoptosis, which are limitations of our study.”
- For replicates, the authors mostly used n = 4 technical replicates from one batch of primary neurons obtained from embryos of one pregnant mouse. The authors should repeat key experiments using true biological replicates, e.g. different batches of neurons from multiple mice.
We want to clarify that every single dot on the graphs represents the mean value for 4 technical replicates from a single batch of primary neurons (obtained from one pregnant mouse) and thus represents one true biological replicate.
In the legend of Figure 1, we have made the following clarification:
“Each dot represents one biological replicate corresponding to a different pregnant mouse, showing the mean value of 4 replicates (4 separate wells) from one batch of primary cortical neurons.”
In the legends of the other figures, we have clarified that each dot represents the mean value of 4 replicates for a unique batch of primary cortical neurons, and that each batch is obtained from a separate pregnant mouse.
In addition, we added the following sentences in the Materials and methods section at the end of the paragraph on OGD:
“Each experiment on wild-type neurons was replicated at least four times, meaning that at least four independently obtained batches of primary cortical neurons were used, each obtained from the embryos of a different pregnant mouse. Every biological replicate (corresponding to one dot on the graph) in turn consisted of four technical replicates (four separate wells with cells from the same batch).”
We hope that this will resolve the misunderstanding.
- The authors discussed the potential importance of astrocytes in mediating apoptosis after OGD. Have the authors tried CDK inhibition experiments using more mature cortical cultures or applying conditioned medium from astrocytes?
This is a very interesting suggestion indeed. We have tried the same experiments on more mature cortical neuronal cultures but have unfortunately struggled to maintain healthy neurons past DIV7.
Regarding the use of conditioned medium from astrocytes, we have not tried it, but we have added the suggestion to the appropriate paragraph in the discussion:
“Therefore, an interesting future experiment could consist of adding conditioned medium from astrocytes to the neuronal culture to see whether this affects our results.” (line 262, page 13).
- Since CDK5 does not require T loop activation by CDK7, could this be an explanation of why flavopiridol, which inhibits CDK5 as well as other CDKs, is neuroprotective but CDK7 inhibition is not?
Again, this is a very interesting comment. Cdk5 is indeed the Cdk that has been most often implicated in neurological diseases, including stroke. While it is certainly possible that Cdk5 activation (that is unaffected by Cdk7 inhibition) could be part of the explanation, there are two sources of doubt which have led us not to include this hypothesis in the discussion:
- Roscovitine also inhibits Cdk5 but was also ineffective in preventing OGD-induced neuronal death in our experiments (despite in vivo experiments that previously showed neuroprotective effects of roscovitine in stroke models).
- It has been suggested that Cdk7 can also be responsible for activating Cdk5 (Rosales et al., 2003), although there is a need for additional research to support this claim.
Bibliography
Li, I.H., Ma, K.H., Weng, S.J., Huang, S.S., Liang, C.M., and Huang, Y.S. (2014). Autophagy activation is involved in 3,4-methylenedioxymethamphetamine ('ecstasy')--induced neurotoxicity in cultured cortical neurons. PLoS One 9, e116565.
Marlier, Q., Jibassia, F., Verteneuil, S., Linden, J., Kaldis, P., Meijer, L., Nguyen, L., Vandenbosch, R., and Malgrange, B. (2018). Genetic and pharmacological inhibition of Cdk1 provides neuroprotection towards ischemic neuronal death. Cell Death Discov 4, 43.
Olson, C.M., Liang, Y., Leggett, A., Park, W.D., Li, L., Mills, C.E., Elsarrag, S.Z., Ficarro, S.B., Zhang, T., Duster, R., et al. (2019). Development of a Selective CDK7 Covalent Inhibitor Reveals Predominant Cell-Cycle Phenotype. Cell Chem Biol 26, 792-803 e710.
Rosales, J., Han, B., and Lee, K.Y. (2003). Cdk7 functions as a cdk5 activating kinase in brain. Cell Physiol Biochem 13, 285-296.
Wu, J.R., Wang, J., Zhou, S.K., Yang, L., Yin, J.L., Cao, J.P., and Cheng, Y.B. (2015). Necrostatin-1 protection of dopaminergic neurons. Neural Regen Res 10, 1120-1124.
Yu, Y., Ren, Q.G., Zhang, Z.H., Zhou, K., Yu, Z.Y., Luo, X., and Wang, W. (2012). Phospho-Rb mediating cell cycle reentry induces early apoptosis following oxygen-glucose deprivation in rat cortical neurons. Neurochem Res 37, 503-511.
Reviewer 2 Report
The article of Tine D'aes et al. is well written, and brings up a very important issue concerning all pre-clinical in vitro stroke studies. The experiments are meticulously conducted, and they provide sufficient evidence of absence of apoptosis in the OGD model of neuronal monoculture. This caveat of the said model is important to bring to the knowledge of everyone involved in preclinical stroke studies, which is why the acceptance of this paper is recommended. The only suggested improvement is to present the results as mean +/- SD instead of SEM.
Author Response
We thank the reviewer for these comments and for drawing our attention to the visual representation of our results. While looking into this, we concluded that, given that we have used nonparametric testing, it would be more correct to represent the data as median +/- interquartile ranges rather than showing the means. We have thus adapted the graphs and the corresponding text in this way.
Reviewer 3 Report
The manuscript Reevaluating the relevance of the oxygen-glucose deprivation model in ischemic stroke: The example of Cdk inhibition focuses on a very important and interesting aspect of research of brain ischemia/reperfusion injury i.e., the appropriateness of OGD models concerning real-life stroke and its relevance to reperfusion phase during ischemic stroke. Moreover, authors’ profound research on effect of inhibition of Cdks in OGD alongside the pharmacological inhibition of autophagy and necroptosis provided valuable data on type of cell death following performed OGD. Special commendation for Section Discussion which is well-written and included excellent explanations of obtained data with biological relevance as well as limitation of conducted research. I have some suggestions/corrections for authors which I believe would improve the quality of the manuscript therefore I strongly recommend their implementation.
List of Suggestions
Since authors have excluded the apoptosis, autophagy, and necroptosis as types of cell death following their OGD model, I would recommend measuring the marker(s) of necrotic cell death such as the miR-122 levels or other to provide evidence for necrosis or at least to include this in discussion as limitation of the study or future perspectives and research.
I would also recommend the authors to exclude the paragraph (lines: 293-303) related to overcoming the flaws of OGD as a model of ischemic stroke, since this is beyond the manuscript’s topic and research conducted.
Please revise/rewrite the following sentence in Section Discussion to sound more clear: To conclude, the unexpected results we observed in our model in relation to the abun- 359 dant literature using OGD to model ischemic stroke suggest that this model should be 360 urgently adapted to model as closely as possible the delayed neuronal death, whether it 361 is truly apoptotic or not, occurring in the penumbra (lines 359-362).
Please specify which fraction is used for Western blot analysis, Section Materials and Methods (Homogenized cells in extraction 442 buffer were left on ice for 20 minutes and centrifuged for 15 minutes (15,000g) at 4°C, lines 442-443).
Author Response
reviewer 3
The manuscript Reevaluating the relevance of the oxygen-glucose deprivation model in ischemic stroke: The example of Cdk inhibition focuses on a very important and interesting aspect of research of brain ischemia/reperfusion injury i.e., the appropriateness of OGD models concerning real-life stroke and its relevance to reperfusion phase during ischemic stroke. Moreover, authors’ profound research on effect of inhibition of Cdks in OGD alongside the pharmacological inhibition of autophagy and necroptosis provided valuable data on type of cell death following performed OGD. Special commendation for Section Discussion which is well-written and included excellent explanations of obtained data with biological relevance as well as limitation of conducted research. I have some suggestions/corrections for authors which I believe would improve the quality of the manuscript therefore I strongly recommend their implementation.
List of Suggestions
Since authors have excluded the apoptosis, autophagy, and necroptosis as types of cell death following their OGD model, I would recommend measuring the marker(s) of necrotic cell death such as the miR-122 levels or other to provide evidence for necrosis or at least to include this in discussion as limitation of the study or future perspectives and research.
We thank the reviewer for this valuable comment, and we agree that this is a limitation of our study, even if we would like to argue that the focus of our experiments was on apoptosis due to the apoptotic nature of Cdk-related neuronal death. We have added the following sentences to the discussion:
“We did not investigate whether neurons expressed markers of autophagy and necroptosis, nor did we confirm the efficacy of the autophagy and necroptosis inhibitors as we did for apoptosis, which are limitations of our study.”
“Future studies should confirm the necrotic nature of this OGD model by measuring the levels of necrotic cell death markers such as the miR-122 (Yang et al., 2014)"
I would also recommend the authors to exclude the paragraph (lines: 293-303) related to overcoming the flaws of OGD as a model of ischemic stroke, since this is beyond the manuscript’s topic and research conducted.
We have removed this paragraph from the discussion as requested.
Please revise/rewrite the following sentence in Section Discussion to sound more clear: To conclude, the unexpected results we observed in our model in relation to the abundant literature using OGD to model ischemic stroke suggest that this model should be 360 urgently adapted to model as closely as possible the delayed neuronal death, whether it 361 is truly apoptotic or not, occurring in the penumbra (lines 359-362).
We agree that this sentence lacked clarity and have rewritten it as follows:
“To conclude, our results observed with the OGD model contribute to the incongruent literature on the mode of cell death that is responsible for OGD-induced neuronal death. They suggest that this model should be adapted to be more reproducible and to better model delayed neuronal death as it occurs in the ischemic penumbra, whether this is truly through apoptotic pathways or not.”
Please specify which fraction is used for Western blot analysis, Section Materials and Methods (Homogenized cells in extraction 442 buffer were left on ice for 20 minutes and centrifuged for 15 minutes (15,000g) at 4°C, lines 442-443).
We have clarified in the text that these were whole cell lysates:
“Homogenized cells in extraction buffer were left on ice for 20 minutes and centrifuged for 15 minutes (15,000g) at 4°C to obtain whole cell lysates.”
Bibliography
Yang, M., Antoine, D.J., Weemhoff, J.L., Jenkins, R.E., Farhood, A., Park, B.K., and Jaeschke, H. (2014). Biomarkers distinguish apoptotic and necrotic cell death during hepatic ischemia/reperfusion injury in mice. Liver Transpl 20, 1372-1382.
Round 2
Reviewer 1 Report
The authors have satisfied my original comments in their response and changes to the manuscript.